# Fast Learning of Multidimensional Hawkes Processes via Frank-Wolfe

**Renbo Zhao**[1*]    **Niccolò Dalmasso**[2]    **Mohsen Ghassemi**[2]
**Vamsi K. Potluru**[2]    **Tucker Balch**[2]    **Manuela Veloso**[2]
[1]Operations Research Center, Massachusetts Institute of Technology
[2]JP Morgan AI Research, New York, NY
`renboz@mit.edu`
`{niccolo.dalmasso, mohsen.ghassemi, vamsi.k.potluru}@jpmchase.com`

## Abstract

Hawkes processes have recently risen to the forefront of tools when it comes to modeling and generating sequential events data. Multidimensional Hawkes processes model both the self and cross-excitation between different types of events and have been applied successfully in various domain such as finance, epidemiology and personalized recommendations, among others. In this work we present an adaptation of the Frank-Wolfe algorithm for learning multidimensional Hawkes processes. Experimental results show that our approach has better or on par accuracy in terms of parameter estimation than other first order methods, while enjoying a significantly faster runtime.

## 1 Introduction

Sequential events data is ubiquitous in many domains of application. In banking, "customer journeys" log the interactions of customer with the bank. In finance, limit order books record buy or sell orders on a security at a specific price or better. In epidemiology, infection data are used to understand the spread pattern of different infectious diseases. Click-stream data in advertising, earthquake magnitude logs for a specific region in seismology, social media posting and interactions, and occurrence if crime in a neighborhood are among other examples of event series data. A *sequence* in event series data is composed of multiple events, potentially of different *types*. Unlike time series data: (a) these event may occur at irregular times, that is the time interval between two events is not predetermined, and (b) event sequences in a sequential event data set need not to be synchronous. Moreover, the probability of occurrence of an event at time $t$ may depend on the *history* of the events in that sequence up to time $t$. Therefore, some of the interesting problems that arise when dealing with sequential events data include predicting the time and the type of the next event, causal inference among the events [1], and intervention to influence the occurrence of future events [2].

A powerful tool to model the occurrence of events in sequential events data is temporal point processes [3]. They model the probability of occurrence of an event of a given type at any time $t$ (possibly as a function of the history up to time $t$). Notably, Hawkes processes (self-exciting point processes) are a common class of temporal point processes [4, 5] that model sequential event series in which occurrence of an event may increase the probability of occurrence of the future events. Temporal point processes are particularly useful as they can also be employed as generative models. Once the temporal process parameters are learned, one can generate synthetic sequential events data with the same temporal dynamics (see, e.g., [6, 7] for simulating data with Hawkes processes dynamics and [8] to do so in a differentially private manner).

---

*Work done during an internship at JP Morgan AI Research.

NeurIPS 2022 Workshop on Synthetic Data for Empowering ML Research.

In this work we focus on efficiently learning multidimensional Hawkes processes, as accurate, fast and scalable optimization algorithms to enable a more widespread use of synthetic sequential events data that reflect dynamics observed in the real data. While in recent years a lot of focus has been devoted to Hawkes process variants that can capture more complex influence patters among events [9, 10, 11, 12], efficient and scalable learning of Hawkes processes parameters have remained relatively under-examined. Especially, when a large number of event types is present in the data, algorithms which are naturally able to provide sparse solutions (i.e., identify that some event types may only influence the occurrence of a (small) subset of other events types in the future) in a fast runtime are needed. Current state-of-the-art approaches achieve that, but either provide no guarantees that solutions found belong to the feasible set [13] or assume upper bounds on dual variables are known and available a priori [14]. In this work, we propose to use a recently proposed first-order optimization algorithm called *Away-Step Frank-Wolfe* for learning multidimensional Hawkes processes. In Section 2.2 we show that the $\ell_1$-regularized maximum likelihood estimation objective can be reformulated as a logarithmic barrier problem over a simplex, which is a natural fit for Frank-Wolfe optimization algorithms. Our numerical experiment show that ASFW is computationally more efficient than state-of-the-art algorithms in solving the $\ell_1$-regularized sparse Hawkes process inference problem.

## 2 Inference of Multidimensional Hawkes Processes

This section provides background on maximum likelihood inference of multidimensional Hawkes processes and details on how we reformulate the optimization problem to be solved efficiently via first order methods.

### 2.1 Maximum-Likelihood Estimation (MLE)

An $m$-dimensional multivariate Hawkes process can be described as follows. Fix a time interval $\mathcal{T} := [0, t)$, and let $\mathcal{D} := \{(t_i, h_i)\}_{i=1}^n$ be the arrival points in this interval, where for each $i \in [n]$, $t_i$ denotes the arrival time and $h_i \in [m]$ denotes the index of the dimension along which the point arrives. For each dimension $k \in [m]$, the conditional density function is given by

$$\lambda_k(t) := \mu_k + \sum_{i:t_i < t} a_{h_i,k} \zeta(t - t_i), \quad \forall t > 0, \tag{1}$$

where $\mu_k \geq 0$ is the base intensity, $a_{h_i,k} \geq 0$ is the mutual-excitation coefficient from dimension $h_i$ to dimension $k$, and $\zeta : \mathbb{R}_+ \to \mathbb{R}_+$ is the kernel function. Define the base intensity vector $\mu := (\mu_k)_{k \in [m]}$ and the cross-activation matrix $A := (a_{k,l})_{k,l \in [m]}$. Given the arrival points $\mathcal{D}$ in $\mathcal{T}$, the log-likelihood function can be written as

$$\begin{aligned}
L(\mu, A) &:= \sum_{i=1}^n \ln \lambda_{h_i}(t_i) - \sum_{k=1}^m \int_0^t \lambda_k(s) \mathrm{d}s \\
&= \sum_{i=1}^n \ln \left( \mu_{h_i} + \sum_{j:t_j < t_i} a_{h_j,h_i} \zeta(-(t_i - t_j)) \right) \\
&\quad - \sum_{k=1}^m \left( \mu_k t + \sum_{i=1}^n a_{h_i,k} \int_0^t \zeta(-(s - t_i)) \mathrm{d}s \right).
\end{aligned} \tag{2}$$

Now, let us define $\mathcal{H}_k(t) := \{i \in [n] : t_i < t, h_i = k\}$ for $k \in [m]$, so that $\{\mathcal{H}_k(t)\}_{k \in [m]}$ form a partition of $[n]$. We assume that $\mathcal{H}_k(t) \neq \emptyset$ for all $k \in [m]$. (This happens with high probability if $t$ is large enough.) We then observe that the log-likelihood function $L(\cdot, \cdot)$ is separable across the $m$ dimensions, namely $L(\mu, A) := \sum_{k=1}^m L_k(\mu_k, a_k)$, where $a_k^\top$ denotes the $k$-th row of $A$ and

$$\begin{aligned}
L_k(\mu_k, a_k) &= \sum_{i \in \mathcal{H}_k(t)} \ln \left( \mu_k + \sum_{l=1}^m a_{l,k} \sum_{j \in \mathcal{H}_l(t_i)} \zeta(-(t_i - t_j)) \right) \\
&\quad - \left( \mu_k t + \sum_{l=1}^m a_{l,k} \sum_{i \in \mathcal{H}_l(t)} \int_0^t \zeta(-(s - t_i)) \mathrm{d}s \right).
\end{aligned} \tag{3}$$

In particular, if we let $\zeta(t) = \exp(-t)$ for $t > 0$ and $0$ for $t \leq 0$, then we have

$$\begin{aligned}
L_k(\mu_k, a_k) &= \sum_{i \in \mathcal{H}_k(t)} \ln \left( \mu_k + \sum_{l=1}^m a_{l,k} \sum_{j \in \mathcal{H}_l(t_i)} \exp(-(t_i - t_j)) \right) \\
&\quad - \left( \mu_k t + \sum_{l=1}^m a_{l,k} \sum_{i \in \mathcal{H}_l(t)} (1 - \exp(-(t - t_i))) \right) \\
&= \sum_{i \in \mathcal{H}_k(t)} \ln \left( \mu_k + \sum_{l=1}^m a_{l,k} \bar{w}_{i,l} \right) - \left( \mu_k t + \sum_{l=1}^m a_{l,k} \bar{v}_l \right),
\end{aligned} \tag{4}$$

where for all $i \in \mathcal{H}_k(t)$ and $l \in [m]$,

$$\bar{w}_{i,l} := \sum_{j \in \mathcal{H}_l(t_i)} \exp(-(t_i - t_j)) \geq 0 \quad \text{and} \quad \bar{v}_l := \sum_{i \in \mathcal{H}_l(t)} (1 - \exp(-(t - t_i))). \tag{5}$$

Note that since $\mathcal{H}_l(t) \neq \emptyset$, we have $\bar{v}_l > 0$, for all $l \in [m]$. Therefore, the maximum-likelihood estimation along the $k$-th dimension is equal to

$$\min_{\mu_k \geq 0, \, a_k \geq 0} \; -\sum_{i \in \mathcal{H}_k(t)} \ln\left(\mu_k + \bar{w}_i^\top a_k\right) + (\mu_k t + \bar{v}^\top a_k). \tag{6}$$

Sometimes, to promote the sparsity of the coefficients $\{a_{l,k}\}_{l=1}^m$, we add a $\ell_1$-regularizer to (6):

$$\min_{\mu_k \geq 0, \, a_k \geq 0} \; -\sum_{i \in \mathcal{H}_k(t)} \ln\left(\mu_k + \bar{w}_i^\top a_k\right) + (\mu_k t + \bar{v}^\top a_k) + \lambda \|a_k\|_1. \tag{7}$$

Note that since $a_k \geq 0$, we can replace $\|a_k\|_1$ with the linear function $e^\top a_k$ (where $e := (1, \ldots, 1)$). The formulation in (7) can also be interpreted as maximum a-posteriori estimation of $a_k$ with independent exponential prior on the elements $\{a_{l,k}\}_{l=1}^m$.

## 2.2 Reformulation

Note that the optimization problems in (6) and (7) both fall under the following optimization model:

$$\bar{f}^* := \min_{z \geq 0, \, z \in \mathbb{R}^q} \left[\bar{f}(z) := -\sum_{i=1}^p \ln\left(w_i^\top z\right) + v^\top z\right]. \tag{8}$$

For example, in (7), we have $p = |\mathcal{H}_k(t)|$,

$$z = \begin{bmatrix} \mu_k \\ a_k \end{bmatrix}, \quad w_i = \begin{bmatrix} 1 \\ \bar{w}_i \end{bmatrix}, \quad v = \begin{bmatrix} t \\ \bar{v} + \lambda e \end{bmatrix}. \tag{9}$$

Since $v > 0$, we can let $y = v \cdot z$ (where $\cdot$ denotes element-wise product) and rewrite (8) as

$$\min_{y \geq 0} \; -\sum_{i=1}^p \ln\left(\widetilde{w}_i^\top y\right) + e^\top y, \tag{10}$$

where $\widetilde{w}_i = w/v$. (Henceforth, for two vectors $x, y \in \mathbb{R}^q$, $x/y$ is interpreted element-wise.) Using standard techniques, namely by writing $y = tx$ for $t \geq 0$ and $x \in \Delta_q := \{x \in \mathbb{R}^q : x \geq 0, e^\top x = 1\}$ and minimizing over $t \geq 0$, we can rewrite (10) as

$$f^* := \min_{x \in \Delta_q} \left[f(x) := -\sum_{i=1}^p \ln\left(\widetilde{w}_i^\top x\right)\right]. \tag{11}$$

We know that $x^*$ is an optimal solution of (11) if and only if $z^* = px^*/v$ is an optimal solution of (8). Note that the objective functions in (8) and (11) have are similar, namely they are both convex but are neither Lipschitz nor have Lipschitz gradient on the feasible set — this poses great challenges to the classical first-order methods (see e.g., [15, 16]). However, compared to the problem in (8), the problem in (11) has a form that is more amenable to some recently developed first-order methods (see Section 3). Due to this, we can solve (11) as a way of solving (8).

## 3 First order methods algorithms for solving (11)

As mentioned in Section 2.2, reformulating the minimization of the log-likelihood as the minimization in (11) allows us to use a series of recently developed first-order methods. More specifically, we consider:

- Multiplicative gradient method [17, 18] (MG).
- Relatively-smooth gradient method (RSGM) [19, 20]: We consider RSGM with fixed step-size (RSGM-F) as proposed in its original development [19, 20] and its variant where the step-size is chosen via backtracking line-search (RSGM-LS) [21] .
- Frank-Wolfe method for log-homogeneous self-concordant barriers (FW-LHB) [22]: We consider FW-LHB with adaptive step-size (FW-LHB-A) and step-size chosen via exact line-search (FW-LHB-E).
- Hawkes ADM4 algorithm for sparse Hawkes processes estimation from the Python library tick [23], which is based on the ADMM algorithm proposed in [13]. We consider the setup with an $\ell_1$ penalty on the infectivity matrix, in order to promote sparsity of the solution.

In addition to those, in this work we present an adaptation of the Frank-Wolfe algorithm [24] to solve (11). The "away-step" addition to the Frank-Wolfe algorithm [25] allows to set one of the solution dimension to zero during each iteration. More specifically, we adapt the Away-step Frank-Wolfe method for log-homogeneous self-concordant barriers [18, AFW-LHB] with adaptive step-size

(`AFW-LHB-A`) and step-size chosen via exact line-search (`AFW-LHB-E`). The resulting adaptation is shown in Algorithm 1. The crucial part of Algorithm 1 is that if the "away step" is chosen in Step 3(b) and the step-size $\alpha_k = \bar{\alpha}_k$ in Step 4, then we have $x_{j_k}^{k+1} = 0$. In other words, we naturally land on one of the verteces of the simplex and we produce naturally "sparse" iterates. This is advantageous in two ways: (i) it allows to identify the sparsity pattern of (the rows of) the cross-activation matrix $A$ and (ii) with the sparsity pattern identified correctly, the convergence rate also becomes faster.

---

**Algorithm 1** Away-Step Frank-Wolfe Method for solving (11)

---

**Input**: Starting point $x^0 \in \Delta_q$. Denote the non-zero (positive) indices of $x^0$ by $\mathcal{I}_0 \subseteq [q]$.
**At iteration** $k \in \{0, 1, \ldots\}$:

1. Compute $i_k \in \arg\min_{i \in [q]} \nabla_i f(x^k)$ and $G_k := \langle \nabla f(x^k), x^k - e_{i_k} \rangle$, where $e_{i_k}$ denotes the $i_k$-th standard coordinate vector.

2. Compute $j_k \in \arg\max_{i \in \mathcal{I}_k} \nabla_i f(x^k)$ and $\widetilde{G}_k := \langle \nabla f(x^k), e_{j_k} - x^k \rangle$.

3. Choose between the following two cases:

   (a) ("Toward step") If $|\mathcal{I}_k| = 1$ or $G_k > \widetilde{G}_k$, let $d^k := e_{i_k} - x^k$ and $\bar{\alpha}_k := 1$.

   (b) ("Away step") Otherwise, let $d^k := x^k - e_{j_k}$ and $\bar{\alpha}_k := x_{j_k}^k / (1 - x_{j_k}^k)$.

4. Choose $\alpha_k \in (0, \bar{\alpha}_k]$ in one of the following two ways:

   (a) Exact line search: $\alpha_k \in \arg\min_{\alpha \in (0, \bar{\alpha}_k]} F(x^k + \alpha d^k)$.

   (b) Adaptive stepsize: Compute $D_k := \langle \nabla^2 f(x^k) d^k, d^k \rangle^{1/2}$ and $r_k := \max\{G_k, \widetilde{G}_k\}$. If $D_k = 0$, then $\alpha_k := \bar{\alpha}_k$; otherwise, $\alpha_k := \min\{b_k, \bar{\alpha}_k\}$, where $b_k := \frac{r_k}{D_k(r_k + D_k)}$.

5. Update $x^{k+1} := x^k + \alpha_k d^k$ and the non-zero (positive) indices of $x^{k+1}$ by $\mathcal{I}_{k+1} \subseteq [q]$.

---

Finally, we also include the Hawkes ADM4 algorithm for sparse Hawkes processes estimation from the Python library `tick` [23], which is based on the ADMM algorithm proposed in [13]. We consider the setup with an $\ell_1$ penalty on the infectivity matrix, in order to promote sparsity of the solution.

## 4 Experimental Results

In this section we report the results in learning multidimensional Hawkes processes both in terms of accuracy in parameters estimation, measured with the $\ell_1$ distance from estimate and ground truth, as well as runtime. We report the results along the first dimension (i.e., $k = 1$), as each dimension can be solved separately (see Section 2.1). We also include goodness of fit measures (log-likelihoods and sub-optimality gap for (11)) in Appendix A. For all experiments, we simulate events data using the code in [26] and choose $x^0 = (m + 1)^{-1} e$ as starting point. Each algorithm generates a sequence of iterates $\{x^t\}_{t \geq 0}$, from which we can recover a sequence of iterates $\{\theta^t\}_{t \geq 0}$ corresponding to (8), where $\theta^t = p x^t / v$ (see (9), with the only exception to the above being the ADMM algorithm from `tick`). All algorithms are run for a maximum of 200 iterations or until converged; since all the methods under comparison are first-order in nature, the computational cost for each method is similar, and hence we use iterations as the horizontal axes for all plots.

### 4.1 Simulated data with $m = 3$ events

We choose the dimension $m = 3$ and time horizon $t = 10^4$. For each dimension $k \in [m]$, we set the ground-truth base-rate $\mu_k^{\text{gt}} = 0.1$. For the ground-truth infectivity matrix $A^{\text{gt}}$, we first generate $A$ with each entry independently drawn from $\mathsf{N}(0.5, 0.2)$, and then projected onto the interval $[0.1, 0.9]$. We set $A^{\text{gt}} = (A + A^\top)/2$, and then randomly choose a pair $(k, l) \in [m]^2$ with $k \neq l$ and set $A_{k,l}^{\text{gt}} = A_{l,k}^{\text{gt}} = 0$. Define the ground-truth parameter along dimension $k$ as $\theta_k^{\text{gt}} := [\mu_k^{\text{gt}}, a_k^{\text{gt}}]$, where $a_k^{\text{gt}}$ denotes the $k$-th row of $A^{\text{gt}}$.

The results are shown in Figure 1. Figure 1, left, shows that `ASFW-LHB-A` and `TICK`) are the fastest algorithm to achieve convergence, shortly followed by `AFW-LHB-E`. (Note that the distance $\|\theta^t - \theta_1^{\text{gt}}\|_1$ does not reduce to zero due to the statistical estimation error caused by the MLE procedure in (6).) This is confirmed by Table 1, where we list the estimated parameter $\hat{\theta}_1 := (\hat{\mu}_1, \hat{a}_1)$ returned by each algorithm after 200 iterations or convergence. Indeed, we see that within very few iterations,

AFW-LHB-type methods can successfully "discover" the sparsity pattern of the ground-truth $\theta_1^{\text{gt}}$, which is a valuable feature by itself. Geometrically, this means AFW-LHB-type methods can quickly drive the iterates $\{\theta^t\}_{t \geq 0}$ to the face of $\Delta_4$ that contains $\theta_1^{\text{gt}}$, which explains their fast convergence. Figure 1, right, shows that the runtime necessary to TICK to reach the same level of accuracy as ASFW-LHB-A and ASFW-LHB-E is around an order of magnitude larger. As we can identify from Table 2, which reports the estimation errors and runtimes averaged over 10 different runs, ASFW-LHB-A averages a runtime of 0.02 seconds against and average of 0.74 seconds for TICK. Note that this result is particularly relevant since ASFW-LHB-A and ASFW-LHB-E have been implemented in Python, while TICK uses an optimized C++ implementation. In terms of goodness of fit, most method achieve a similar log-likelihood and ASFW-LHB-A and AFW-LHB-E achieve sub-linear convergence (see Appendix A); this observation indeed can be justified from theoretical convergence results [22, 27, 18].

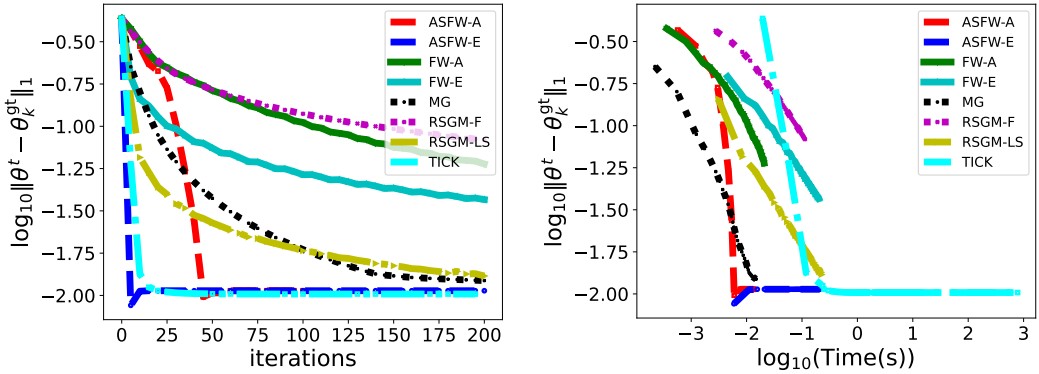

Figure 1: Comparison of different algorithms stated in Section 3 in terms of (a) sub-optimality gap of (11) and (b) distance to the ground-truth parameter $\theta_1^{\text{gt}}$. ASFW-LHB-A, ASFW-LHB-E and TICK achieve the same accuracy in estimating the Hawkes process parameters, but the first two algorithms are an order of magnitude faster in terms of runtime.

Table 1: The estimated $\mu_1$ and $a_1$ returned by each algorithm in Section 3 after 200 iterations or convergence. Note ASFW-LHB-A and ASFW-LHB-E are the only algorithm to return exactly 0 for $\widehat{a}_1^2$, which is a consequence of the away-step (see Section 3.)

| Algorithm | $\hat{\theta}_1 := (\hat{\mu}_1, \widehat{a}_1)$ |
|---|---|
| ASFW-LHB-A | (0.101 , 0.298, 0.000, 0.281) |
| ASFW-LHB-E | (0.101 , 0.298, 0.000, 0.281) |
| FW-LHB-A | (0.097 , 0.297, 0.036, 0.264) |
| FW-LHB-E | (0.098 , 0.297, 0.023, 0.272) |
| MG | (0.100 , 0.298, 0.007, 0.280) |
| RSGM-LS | (0.100 , 0.298, 0.009, 0.279) |
| RSGM-F | (0.065 , 0.295, 0.050, 0.257) |
| TICK | (0.101 , 0.299, 0.003, 0.282) |
| Ground-truth | (0.100 , 0.302, 0.000, 0.279) |

## 4.2 Extending to higher $m$ and different sparsity patterns

In this section we extend the simulation results to higher number of events $m$, as well as experimenting with different levels of sparsity in the cross-activation matrix $A$. More specifically, we simulate events in the same settings as Section 4.1 with $m = 5$ and $m = 10$, including three levels of sparsity: 30%, 50% and 70%. For $m = 5$ we use a time horizon $t = 5 \times 10^4$ and for $m = 10$, $t = 10^5$; all results are averaged over 10 separate runs. Table 3 reports that the estimation errors for ASFW-LHB-A, ASFW-LHB-E and TICK is comparable across different levels of sparsity and number of events; see Table 4 in Appendix A for a full comparison with all algorithms presented in Section 3, showing that ASFW-LHB-A, ASFW-LHB-E and TICK are consistently the most accurate. Figure 2 shows that the faster runtime for ASFW-LHB-A and ASFW-LHB-E extends to multiple number of events, across

Table 2: Estimation error $\mu_1$, $a_1$ and overall $\theta_1 = (\mu_1, a_1)$, along with runtime, for $m = 3$, averaged over 10 runs.

| Algorithm | $\|\widehat{\mu_1} - \mu_1\|$ $(10^{-2})$ | $\|\widehat{a_1} - a_1\|_1$ $(10^{-2})$ | Runtime $(s)$ | $\|\widehat{\theta_1} - \theta_1\|_1$ $(10^{-2})$ |
|---|---|---|---|---|
| ASFW-LHB-A | $0.264 \pm 0.125$ | $\mathbf{2.627 \pm 1.149}$ | $\mathbf{0.02 \pm 0.01}$ | $\mathbf{2.036 \pm 0.869}$ |
| ASFW-LHB-E | $0.264 \pm 0.125$ | $\mathbf{2.627 \pm 1.149}$ | $0.27 \pm 0.21$ | $\mathbf{2.036 \pm 0.869}$ |
| FW-LHB-A | $0.904 \pm 0.151$ | $7.803 \pm 1.900$ | $0.03 \pm 0.01$ | $6.078 \pm 1.449$ |
| FW-LHB-E | $0.555 \pm 0.116$ | $4.531 \pm 1.234$ | $0.34 \pm 0.15$ | $3.537 \pm 0.935$ |
| MG | $0.270 \pm 0.124$ | $2.665 \pm 1.140$ | $\mathbf{0.02 \pm 0.01}$ | $2.066 \pm 0.862$ |
| RSGM-LS | $0.910 \pm 0.094$ | $7.225 \pm 1.626$ | $0.19 \pm 0.06$ | $5.646 \pm 1.227$ |
| RSGM-F | $0.362 \pm 0.118$ | $3.193 \pm 1.066$ | $0.38 \pm 0.11$ | $2.485 \pm 0.803$ |
| TICK | $\mathbf{0.262 \pm 0.123}$ | $2.644 \pm 1.147$ | $0.74 \pm 0.74$ | $2.049 \pm 0.868$ |

the different sparsity levels. In addition, one can see that both a higher sparsity level and size of the cross-activation matrix require a higher runtime.

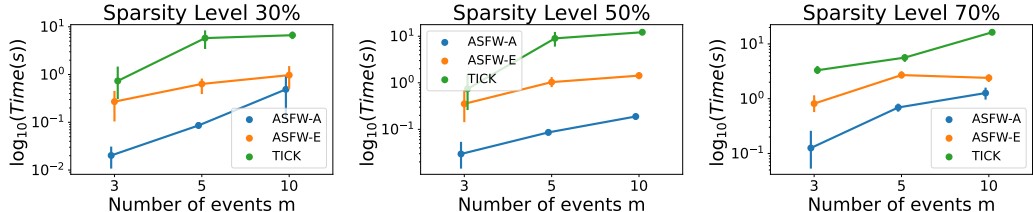

Figure 2: Runtime for the `ASFW-LHB-A`, `ASFW-LHB-E` and `TICK` for different number of events $m$ (x-axis) and across different levels of sparsity of the cross-activation matrix $A$. `ASFW-LHB-A` and `ASFW-LHB-E` remain significantly faster than `TICK` across the board, scaling roughly linearly with size and sparsity level of the cross-activation matrix.

Table 3: Estimation error for $\theta_1 = (\mu_1, a_1)$ across different number of events $m$ and sparsity levels of the cross-activation matrix $A$, averaged across 10 runs. `ASFW-LHB-A`, `ASFW-LHB-E` and `TICK` show essentially identical performance across all settings.

| Sparsity Level | Algorithm | $m = 3$ | $m = 5$ | $m = 10$ |
|---|---|---|---|---|
| 30% | ASFW-LHB-A | $2.036 \pm 0.869$ | $1.403 \pm 0.470$ | $3.093 \pm 0.733$ |
| | ASFW-LHB-E | $2.036 \pm 0.869$ | $1.403 \pm 0.470$ | $3.095 \pm 0.735$ |
| | TICK | $2.049 \pm 0.868$ | $1.403 \pm 0.470$ | $3.104 \pm 0.732$ |
| 50% | ASFW-LHB-A | $2.355 \pm 0.608$ | $1.592 \pm 0.391$ | $3.600 \pm 0.544$ |
| | ASFW-LHB-E | $2.355 \pm 0.608$ | $1.592 \pm 0.391$ | $3.244 \pm 0.728$ |
| | TICK | $2.370 \pm 0.604$ | $1.592 \pm 0.392$ | $3.249 \pm 0.726$ |
| 70% | ASFW-LHB-A | $1.287 \pm 0.331$ | $1.690 \pm 0.358$ | $3.250 \pm 0.410$ |
| | ASFW-LHB-E | $1.287 \pm 0.331$ | $1.690 \pm 0.358$ | $3.250 \pm 0.410$ |
| | TICK | $1.289 \pm 0.332$ | $1.690 \pm 0.358$ | $3.261 \pm 0.407$ |

## 5 Conclusions

Multidimensional Hawkes processes are powerful mathematical tools for modeling and generating sequential events data, with widespread applications in banking, finance, healthcare, seismology, advertising, and so on. In this work, we propose employing the recently developed Away-Step Frank-Wolfe algorithm for scalable learning of MHPs. This proposition is supported by our observation that the maximum likelihood estimation problem in learning MHPs can be posed as a logarithmic barrier function optimization problem over a simplex, making it a natural fit for Frank-Wolfe-type algorithms. Experiment results show that our proposed adaptation with either an adaptive step size `ASFW-LHB-A` or exact line search `ASFW-LHB-E` achieve on par performance with many first orders algorithms, while enjoying a significantly faster runtime. For future work, we would like to explore

downstream benefits in performance for machine learning task when using synthetic events data simulated by more accurately estimated Hawkes processes, as well as implications for differentially private Hawkes processes (as fewer passes over the data are required to achieve convergence).

## Acknowledgments and Disclosure of Funding

This paper was prepared for informational purposes by the Artificial Intelligence Research group of JPMorgan Chase & Co. and its affiliates ("JP Morgan"), and is not a product of the Research Department of JP Morgan. JP Morgan makes no representation and warranty whatsoever and disclaims all liability, for the completeness, accuracy or reliability of the information contained herein. This document is not intended as investment research or investment advice, or a recommendation, offer or solicitation for the purchase or sale of any security, financial instrument, financial product or service, or to be used in any way for evaluating the merits of participating in any transaction, and shall not constitute a solicitation under any jurisdiction or to any person, if such solicitation under such jurisdiction or to such person would be unlawful.

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

# A Experimental Results

We include this section to include all figures and tables which could not be included in Section 4 for a space reason. Figure 3 shows the sub-optimality gap and distance to the log-likelihood achieved by the ground-truth parameter for all first order methods used in Section 4.1. Table 4 shows the estimation errors for all algorithms in the settings of Section 4.2.

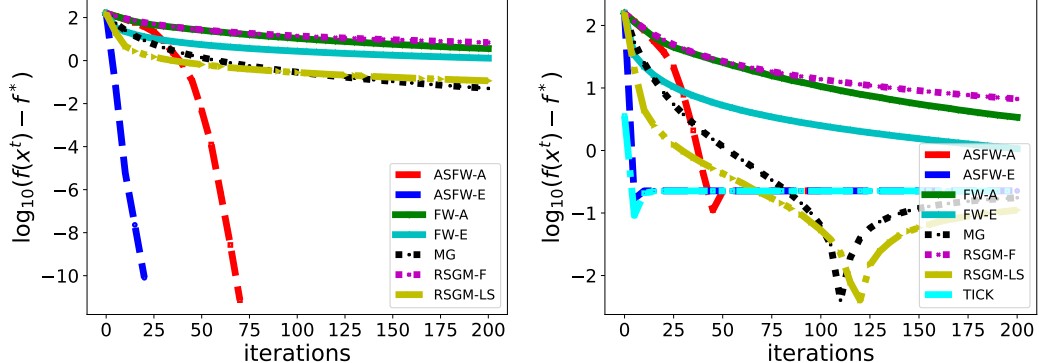

Figure 3: Comparison of algorithms presented in Section 3 in terms of sub-optimality gap of (11) (left) and distance to the log-likelihood achieved by the ground-truth parameter $\theta_1^{\mathrm{gt}}$ (right). Note that the sub-optimality gap is not available in `tick`, so it is missing in the left figure, and that the log-likelihood is not stable due to the statistical errors in simulating sequential events data from [26]

Table 4: Estimation error for $\theta_1 = (\mu_1, a_1)$ across different number of events $m$ and sparsity levels of the cross-activation matrix $A$, for all algorithms presented in Section 3.

| Sparsity Level | Algorithm | $m = 3$ | $m = 5$ | $m = 10$ |
|---|---|---|---|---|
| 30% | ASFW-LHB-A | **2.036 ± 0.869** | **1.403 ± 0.470** | **3.093 ± 0.733** |
| | ASFW-LHB-E | **2.036 ± 0.869** | **1.403 ± 0.470** | 3.095 ± 0.735 |
| | FW-LHB-A | 6.078 ± 1.449 | 9.462 ± 1.702 | 14.768 ± 2.375 |
| | FW-LHB-E | 3.537 ± 0.935 | 3.123 ± 0.707 | 5.252 ± 1.092 |
| | MG | 2.066 ± 0.862 | 1.476 ± 0.468 | 3.297 ± 0.702 |
| | RSGM-F | 2.485 ± 0.803 | 2.307 ± 0.401 | 4.093 ± 0.682 |
| | RSGM-LS | 5.646 ± 1.227 | 7.777 ± 0.969 | 16.767 ± 0.623 |
| | TICK | 2.049 ± 0.868 | **1.403 ± 0.470** | 3.104 ± 0.732 |
| 50% | ASFW-LHB-A | **2.355 ± 0.608** | **1.592 ± 0.391** | 3.600 ± 0.544 |
| | ASFW-LHB-E | **2.355 ± 0.608** | **1.592 ± 0.391** | **3.244 ± 0.728** |
| | FW-LHB-A | 4.052 ± 0.683 | 11.784 ± 1.015 | 15.833 ± 1.395 |
| | FW-LHB-E | 3.055 ± 0.529 | 3.989 ± 0.292 | 6.320 ± 0.633 |
| | MG | 2.373 ± 0.604 | 1.627 ± 0.377 | 3.457 ± 0.695 |
| | RSGM-F | 2.533 ± 0.561 | 2.053 ± 0.273 | 4.084 ± 0.573 |
| | RSGM-LS | 4.016 ± 0.694 | 7.268 ± 0.625 | 15.124 ± 1.708 |
| | TICK | 2.370 ± 0.604 | **1.592 ± 0.392** | **3.249 ± 0.726** |
| 70% | ASFW-LHB-A | **1.287 ± 0.331** | **1.690 ± 0.358** | **3.250 ± 0.410** |
| | ASFW-LHB-E | **1.287 ± 0.331** | **1.690 ± 0.358** | **3.250 ± 0.410** |
| | FW-LHB-A | 9.748 ± 0.857 | 2.171 ± 0.333 | 13.905 ± 2.921 |
| | FW-LHB-E | 3.938 ± 0.429 | **1.690 ± 0.358** | 6.010 ± 0.667 |
| | MG | 1.405 ± 0.359 | 2.309 ± 0.402 | 3.660 ± 0.333 |
| | RSGM-F | 1.693 ± 0.370 | **1.690 ± 0.358** | 3.803 ± 0.305 |
| | RSGM-LS | 6.445 ± 0.471 | 14.092 ± 1.712 | 14.933 ± 2.172 |
| | TICK | **1.289 ± 0.332** | **1.690 ± 0.358** | **3.261 ± 0.407** |

