# OpenReview forum: "Fast Learning of Multidimensional Hawkes Processes via Frank-Wolfe"
_NeurIPS.cc/2022/Workshop/SyntheticData4ML — Neurips 2022 SyntheticData4ML_

### Official Review · Reviewer_9nr3 · 2022-10-15
**Mode details and explanation needed but potentially interesting**

**Rating:** 5
**Confidence:** 3

**Review:**

This paper introduces a potentially useful new algorithm for estimating parameters in a Hawkes process in order to generate synthetic data from it.

Pros: The method appears to be faster but similarly accurate as its main competitors.

Cons: The method is not clearly described and the results are hard to follow. Here are some questions.
* What are the independence assumptions in the Hawkes process? Why is the matrix A taken to be symmetric in the examples, is that a model assumption?
* Are there theoretical guarantees for Algorithm 1? Will it always converge and is there some consistency guarantees?
* The simulated data only have 3 events, and only the results for the first event are shown. Is everything assumed to be independent? What would be the ground truth in the simulations?
* Most algorithms are implemented in Python but TICK is implemented in C++. How much do the runtime results depend on the implementation?
* The paper is motivated by examples such as limit order books but in that example the number of events would be much larger than 3. How does the algorithm scale with the number of events?
* Figure 3 in the SI: why is there a dip for RSGM-F and for MG? The newly proposed methods ASFW_A and ASFW_E also seem to show a dip; is there an explanation? The statement in the legend ``the log-likelihood is not stable due to the statistical errors in simulating sequential events data from [26]'' is not clear. Is this not the same simulated data set as for the main text? How do statistical errors arise in simulating sequential events data?

As an aside,  in Figure 1 the colour for RW-E and TICK is almost the same. Choosing a different colour would be recommended.

---

### Official Review · Reviewer_mScj · 2022-10-16
**Event generation via a novel approach to optimization (excellent paper)**

**Rating:** 7
**Confidence:** 3

**Review:**

1. Summary
The authors reformulate MLE optimization for Hawkes process learning as an optimization over a simplex and apply Frank-Wolfe algorithm modification to solve this optimization problem.


2. Strengths and weaknesses.
Strengths:
- Very good writing and presentation
- Interesting approach
- Important problem for synthetic data generation community

Weaknesses:
- some references are anonymized, so it is hard to evaluate the novelty of this paper (18, 22)
- only toy experiments, no experiments on real data
- no theoretical analysis for convergence rate.

3. Questions
Could you estimate the number of dimensions and events where this method might be practical?

You wrote:
- "However, compared to the problem in (8), the problem in (11) has a form that is more amenable to some recently developed first-order methods (see Section 3)."
- In Section 3, you list the methods. It would be great to add a brief comment with an intuitive explanation of why these methods perform well, and the initial optimization problem is more complicated than your's reformulation.


4. Limitations
I think the limitations are well discussed, except for using exponential kernel (see Issues).


5. Issues.
- The method seem to work only for $\zeta(t)$ exponential. I think you need to put more emphasis on it and discuss the limitations.
- Table 1, IMO, readers are not interested in the exact values of the parameters. They are interested in how far the parameters are from the ground truth.


6. Review Summary
Good paper; the theoretical analysis and experimental results are convincing. **I cannot check novelty fully as some of the references are anonymised.**

---

### Meta-Review · Area_Chair_thzn · 2022-10-18

**Recommendation:** Accept